# Mass cytometry reveals cellular fingerprint associated with IgE+ peanut tolerance and allergy in early life

Melanie R. Neeland [1,2,7✉], Sandra Andorf [3,4,5,7], Monali Manohar[3], Diane Dunham[3], Shu-Chen Lyu[3], Thanh D. Dang[1,2], Rachel L. Peters[1,2], Kirsten P. Perrett[1,6], Mimi L.K. Tang [1,2], Richard Saffery [1,2], Jennifer J. Koplin[1,2,6] & Kari C. Nadeau [3✉]

IgE-mediated peanut allergic is common, often serious, and usually lifelong. Not all individuals who produce peanut-specific IgE will react upon consumption of peanut and can eat the food without adverse reactions, known as sensitized tolerance. Here, we employ high-dimensional mass cytometry to define the circulating immune cell signatures associated with sensitized tolerance and clinical allergy to peanut in the first year of life. Key features of clinical peanut allergic are increased frequency of activated B cells (CD19hiHLADRhi), overproduction of TNFα and increased frequency of peanut-specific memory CD4 T cells. Infants with sensitized tolerance display reduced frequency but hyper-responsive naive CD4 T cells and an increased frequency of plasmacytoid dendritic cells. This work demonstrates the utility and power of high-dimensional mass cytometry analysis to interrogate the cellular interactions that are associated with allergic sensitization and clinical food allergy in the first year of life.

[1] Murdoch Children's Research Institute, Parkville, VIC, Australia. [2] Department of Pediatrics, The University of Melbourne, Parkville, VIC, Australia. [3] Sean N. Parker Center for Allergy and Asthma Research at Stanford University, Stanford, California, USA. [4] Divisions of Biomedical Informatics and Allergy & Immunology, Cincinnati Children's Hospital Medical Center, Cincinnati, Ohio, USA. [5] Department of Pediatrics, University of Cincinnati College of Medicine, Cincinnati, Ohio, USA. [6] School of Population and Global Health, The University of Melbourne, Parkville, VIC, Australia. [7] These authors contributed equally: Melanie R. Neeland, Sandra Andorf. ✉email: melanie.neeland@mcri.edu.au; knadeau@stanford.edu

gE-mediated food allergy is one of the earliest manifestations of allergic disease and is a major public health concern, affecting ~10% of infants and 5% of school-aged children[1–3]. However, not all individuals who produce allergen-specific IgE (sIgE) will react upon consumption of the offending allergen and can eat the food without adverse consequences, a unique clinical phenotype known as sensitized tolerance[4]. At present, the extent of immune system dysfunction in food allergy and the factors that govern the transition from sensitization to clinical allergy are largely understudied, particularly in the first year of life when food allergy develops and when allergenic foods are first introduced into the diet. In fact, elucidating the cellular mechanisms that account for the differences between innate tolerance, food sensitization, and food allergy was identified as a top research priority by the National Academies of Sciences, Engineering, and Medicine International Expert Panel on Food Allergy in 2017[5].

Unlike other food allergies, such as egg allergy that naturally resolves in up to 90% of children by the age of 6 years, peanut allergic usually persists and is often lifelong[6]. Furthermore, children with peanut and treenut allergies are threefold more likely to have a reaction consistent with anaphylaxis compared with those with other food allergies[2,7]. Currently, peanut avoidance remains the only management strategy; however accidental ingestion is common and can lead to life-threatening reactions. Understanding why some infants with peanut sIgE go on to develop peanut allergic, whereas others do not, is crucial for accurate diagnosis and early intervention. We have previously shown that food allergy in infancy is associated with marked differential immune cell profiles and cytokine production from both the innate and adaptive immune systems[8–11]. However, to date, the complete immune profile and immune system interactions in infants with food sensitization and food allergy remains to be investigated.

In the present study, we use high-dimensional mass cytometry to conduct a comprehensive phenotypic and functional investigation of immune parameters in a group of well-characterized 1-year-old infants with clinical peanut allergic (as defined by positive food challenge), infants with peanut-sensitized tolerance (as defined by negative food challenge but the presence of peanut sIgE), and non-sensitized non-allergic healthy control infants following in-vitro antigen-specific (peanut) and nonspecific (Phorbal myristate acetate (PMA)/ionomycin) stimulation. A combination of high-dimensional, unsupervised computational analyses and manual gating is used to comprehensively explore the immune signatures associated with clinical allergy and tolerance to peanut in the first year of life.

## Results

**Subjects and study design**. A subset of thirty-six 1-year-old infants from the population-based HealthNuts cohort ($n = 5276$)[12] were selected for this study (comprising $n = 12$ each of challenge-confirmed peanut-allergic (PA) infants, peanut-sensitized tolerant (PST) infants, and non-sensitized, non-food-allergic (NA) infants) (Fig. 1). Oral food challenges (OFCs) were performed in all subjects according to standardized protocols, as detailed in the Methods section, to confirm peanut allergic status. PA infants were defined as having a skin prick test (SPT) wheal diameter ≥2 mm or an sIgE level of ≥0.35 kUA/L to peanut, and an unequivocal objective allergic reaction during peanut OFC at age 1 year. PST infants had a positive SPT ≥2 mm and sIgE level of ≥0.35 kUA/L to peanut and a negative peanut OFC at age 1 year. Healthy control infants (the NA group) were non-sensitized and non-allergic, with a negative SPT to peanut, egg, sesame, and cow's milk together with a negative peanut OFC outcome at age 1 year. Table 1 describes the demographics and clinical characteristics of the three groups. The PST and PA infants showed greater peanut sIgE and SPT values than any of the NA infants. Although the peanut sIgE levels were comparable between the PST and PA infants, the PST participants had on average smaller SPT values than the PA infants (Supplementary Fig. 1).

**Mass cytometry as a tool for immune profiling in infants**. We first determined the cellular phenotype and frequencies obtained by mass cytometry in unstimulated cells within the NA healthy infants ($n = 12$). Table 2 and Fig. 2a show the results of manual gating analysis for the major cell populations: CD4 T cells, CD8 T cells, B cells, narural killer (NK) cells, monocytes, and dendritic cells (DCs) (as percentage of live cells). The subsequent analysis of all 36 infants, stratified by clinical outcome, in this study is shown in Fig. 2b. A summary of sub-typing of the major populations is reported in Table 2 and in Fig. 2c.

Nonspecific stimulation of peripheral blood mononuclear cells (PBMCs) with PMA/ionomycin and assessment of intracellular cytokine production were employed to assess the capacity of cells to respond to activation. Each of NK, CD4 T, CD8 T, B cells, monocytes, and DCs from NA individuals responded to stimulation. Interestingly, NK cells produced the most cytokines of any cell population, with 53.65% (range: 26.2–77.8%) of NK cells producing interferon-γ (IFNγ) and 31.9% (9.24-47.6%) producing tumor necrosis factor-α (TNFα) (Fig. 2d). CD4 and CD8 T cells responded similarly, producing a range of IFNγ, interleukin-2 (IL-2), and TNFα. B cells, monocytes, and DCs produced only TNFα following stimulation (Fig. 2d).

**Immune profiling by unsupervised analysis**. Clustering analysis with FlowSOM[13] using 18 lineage markers revealed 16 cell clusters (Fig. 3a). Based on the expression of lineage markers, the clusters were classified into the following 11 cell types with the following frequencies (% live cells): naive CD4 T cells (CD3+ CD4+ CD45RA+ CCR7+ (37.79%)), effector memory CD4 T cells (CD3+ CD4+ CD45RA− CCR7− HLADR−(5.35%)), HLADR+ effector memory CD4 T cells (CD3+ CD4+ CD45RA−CCR7− HLADR+ (2.36%)), central memory CD4 T cells (CD3+ CD4+ CD45RA− CCR7+ (1.30%)), regulatory T cells (Tregs) (CD3+ CD4+ CD25+ CD127− (2.26%)), naive CD8 T cells (CD3+ CD8+ CD45RA+ CCR7+ (10.81%)), effector memory CD8 T cells (CD3+ CD8+ CD45RA− CCR7− (1.28%)), central memory CD8 T cells (CD3+ CD8+ CD45RA− CCR7+ (1.03%)), B cells (cluster 1, CD3− CD19+ CD20+ HLADR+ (7.23%)), B cells (cluster 2, CD3− CD19++ CD20+ HLADR++ (12.19%)), NK cells (CD3− CD19− CD14− CD56+ (1.8%)), DCs (CD3− CD19− CD56− CD14− HLADR+ (1.88%)), and monocytes (CD3− CD19− CD56− CD14+ HLADR+ CD11c+ (1.62%)). Three clusters were unable to be confidently classified, including undefined CD3+ T cells (two clusters combined; 5.87%, both negative for CD4 and CD8) and a cluster that was negative for all markers (4.3%). These three clusters have been removed from statistical analysis comparing the three study groups. The frequency of each of the clusters identified by unsupervised analysis from the 36 infants in this study is shown in Fig. 3b.

Importantly, the frequencies of the major cell types identified by unsupervised analysis were comparable to those obtained by manual gating (Supplementary Fig. 2A). Specifically, median frequencies (% live cells) across the 36 infants of each major population identified by manual gating vs unsupervised clustering, respectively, were: CD4 T cells (44.53 vs. 46.86), CD8 T cells (12.33 vs. 13.66), B cells (20.07 vs. 19.9), NK cells (1.99 vs. 1.37), monocytes (1.42 vs. 1.36), and DCs (0.89 vs. 0.92). Frequencies of the major cell populations as determined by unsupervised clustering for individual infants are presented in Supplementary Fig. 2B. To further verify our clustering results and to visualize

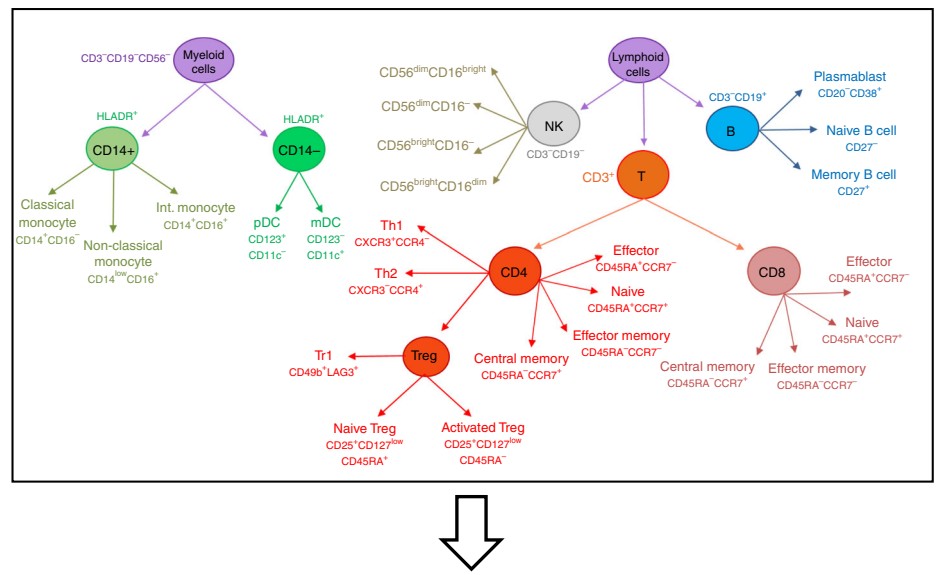

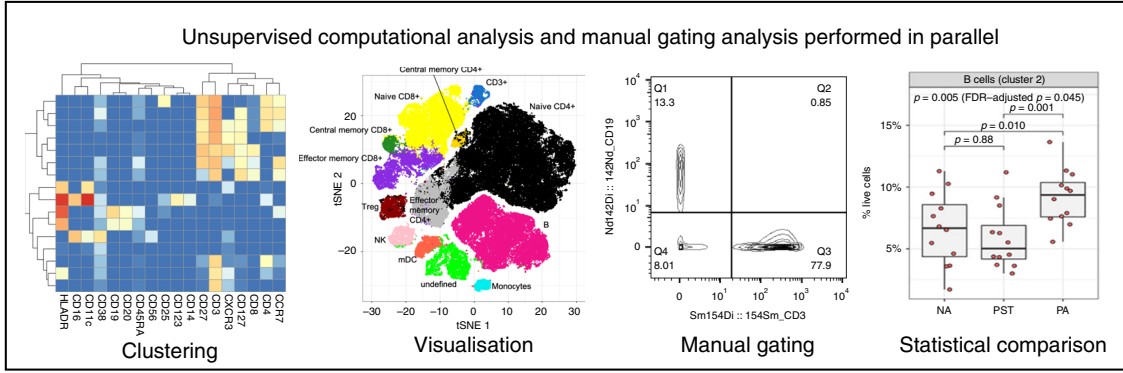

**Fig. 1 Experimental workflow for mass cytometry analysis of PBMCs from 1-year-old infants.** Cryopreserved PBMCs from $n = 12$ peanut-allergic (PA), $n = 12$ peanut-sensitized tolerant (PST), and $n = 12$ non-allergic (NA) healthy 1-year-old infants were thawed and underwent a 24 h stimulation with media (control), pure peanut protein (specific stimulation), or PMA/ionomycin (nonspecific stimulation). Samples were barcoded and stained with a panel of antibodies against 24 surface markers and 8 intracellular markers, and analyzed with a Helios Mass Cytometer. For data analysis, unsupervised computational analyses (clustering and visualization) and manual gating analyses were performed in parallel, along with statistical analyses to identify immune signatures significantly different between the clinical groups.

this high-dimensional data, we performed a Uniform Manifold Approximation and Projection (UMAP) analysis as shown in Fig. 3c. The cells were color highlighted by their respective FlowSOM cluster. CD4 T-cell subpopulations, CD8 T-cell subpopulations, B-cell populations, and the myeloid populations cluster together within two-dimensional space, corresponding well to the automatically defined clusters.

**Immune signatures that discriminate allergy from tolerance.** We next compared resting cell populations between PA infants,

PST infants, and NA controls. Results of our unsupervised clustering, followed by mixed-effects model analysis, revealed significant differences between the three groups in the abundance of two clusters, corresponding to the naive CD4 T-cell cluster ($p = 0.007$, $\chi^2 = 9.9$, false discovery rate (FDR)-adjusted $p = 0.045$, $\chi^2$-test, Fig. 4a) and the $CD19^{++}HLADR^{++}$ B cell (B-cell cluster 2) ($p = 0.005$, $\chi^2 = 10.8$, FDR-adjusted $p = 0.045$, $\chi^2$-test, Fig. 4b). Pairwise post-hoc analyses revealed that PA infants show increases in the frequency of these B cells relative to PST and NA infants ($p = 0.001$, $\chi^2 = 10.2$ and $p = 0.01$, $\chi^2 = 6.6$, respectively, $\chi^2$-test) (Fig. 4b). PST

**Table 1 Demographics and clinical characteristics of study cohort.**

|  | NA | PST | PA | *p*⋆ |
|---|---|---|---|---|
| Total number | 12 | 12 | 12 |  |
| Sex: male, *n* (%) | 5 (42%) | 7 (58%) | 8 (67%) | 0.59 |
| Both parents born in Australia, *n* (%) | 11 (92%) | 8 (67%) | 5 (42%) | 0.04 |
| Family history of allergy[a], *n* (%) | 9 (75%) | 9 (75%) | 8 (67%) | 1 |
| Eczema at age 1 year[b], *n* (%) | 4 (33%) | 6 (50%) | 5 (42%) | 0.91 |
| Peanut SPT (mm), median (IQR) | 0 (0) | 3.25 (1.38) | 9.0 (2.0) | 0.0001** |
| Peanut sIgE (kUA/L)[c], median (IQR) | 0.005 (0.015) [3 ND] | 1.14 (1.24) | 4.24 (10.54) [3 ND] | 0.11** |
| Egg allergic, (%) | 0 (0%) | 9 (75%) | 10 (83%) | <0.0001 (1**) |
| Sesame allergic | 0 (0%) | 0 (0%) | 0 (0%) | 1 |
| Sensitized to cow's milk[d] | 0 (0%) | 1 (8%) | 2 (17%) | 0.45 |
| Sensitized to house dust mite[d] | 0 (0%) | 1 (8%) | 2 (17%) | 0.76 |

*IQR* interquartile range, *ND* data not available.
⋆*p*-values by Fisher's exact test between the three groups.
**p*-values by Fisher's exact test or Wilcoxon's rank-sum test for the comparison between PST and PA.
[a]Asthma, allergic rhinitis, eczema, or food allergy.
[b]Doctor diagnosed eczema requiring topical steroid treatment or eczema observed by a trained nurse.
[c]Values at the lower detection limit of 0.01 kUA/L were set to 0.005.
[d]Skin prick test ≥2 mm.

**Table 2 Summary of cell frequencies (% of parent) by manual gating within the healthy infants (group NA, $n = 12$).**

| Parent population | Cell population | Minimum | 1st Quartile | Median | Mean | 3rd Quartile | Maximum |
|---|---|---|---|---|---|---|---|
| Live cells | CD4 T cells | 39.8 | 41.83 | 45.59 | 47.7 | 50.94 | 64.24 |
|  | CD8 T cells | 3.23 | 11.34 | 13.42 | 12.66 | 14.83 | 19.85 |
|  | B cells | 10.24 | 13.97 | 17.09 | 18.28 | 23.47 | 26.08 |
|  | NK cells | 0.37 | 1.43 | 2.0 | 1.9 | 2.42 | 2.95 |
|  | Monocytes | 0.36 | 1.33 | 1.42 | 1.59 | 1.77 | 3.06 |
|  | DCs | 0.42 | 0.51 | 0.81 | 1.75 | 1.16 | 9.74 |
| CD4 T cells | Naive CD4 T cells | 42.8 | 59.25 | 63.9 | 62.93 | 67.25 | 77.8 |
|  | Central memory CD4 T cells | 13.2 | 17.65 | 20.2 | 20.93 | 22.62 | 34.5 |
|  | Effector memory CD4 T cells | 4.11 | 5.66 | 7.28 | 8.77 | 10.62 | 19.4 |
|  | Effector CD4 T cells | 3.29 | 4.62 | 5.89 | 7.37 | 10.22 | 17.6 |
|  | Regulatory T cells | 2.74 | 3.9 | 4.62 | 4.43 | 5.22 | 5.45 |
| CD8 T cells | Naive CD8 T cells | 58.1 | 63.85 | 68.2 | 70.8 | 79.08 | 84.5 |
|  | Central memory CD8 T cells | 3.99 | 5.31 | 6.39 | 6.64 | 7.5 | 11.9 |
|  | Effector memory CD8 T cells | 3.66 | 6.54 | 12.45 | 10.99 | 13.85 | 19.3 |
|  | Effector CD8 T cells | 5.21 | 6.75 | 12.15 | 11.56 | 13.78 | 20.7 |
| B cells | Naive B cell | 92.0 | 95.57 | 96.15 | 95.97 | 96.82 | 98.8 |
|  | Memory B cells | 1.21 | 3.15 | 3.84 | 4.03 | 4.42 | 8.05 |
| NK cells | CD56+CD16+ NK | 20.2 | 48.17 | 55.7 | 53.19 | 64.75 | 67.2 |
|  | CD56+CD16- NK | 20.3 | 22.77 | 25.3 | 31.82 | 34.45 | 76.7 |
|  | CD56bright CD16+ NK | 0.78 | 2.21 | 2.84 | 3.25 | 3.34 | 8.97 |
|  | CD56bright CD16− NK | 2.34 | 7.3 | 11.65 | 11.76 | 13.65 | 27.7 |
| Monocytes | CD14+CD16+ monocytes | 4.83 | 46.77 | 53.0 | 49.73 | 56.73 | 70.6 |
|  | CD14+CD16− monocytes | 29.4 | 43.27 | 47.0 | 50.27 | 53.23 | 95.2 |
| DCs | mDC % HLADR+ | 13.0 | 24.55 | 30.0 | 36.69 | 44.82 | 80.7 |
|  | pDC % HLADR+ | 0.65 | 1.36 | 2.32 | 2.44 | 3.24 | 5.24 |

infants, however, show reduced frequency of naive CD4 T cells relative to both PA infants ($p = 0.043$, $\chi^2 = 4.1$, $\chi^2$-test) and NA controls ($p = 0.006$, $\chi^2 = 7.6$, $\chi^2$-test) (Fig. 4a). The results of this unsupervised analysis were confirmed by manual gating (Supplementary Fig. 3A, B). Rare populations of interest not detected by unsupervised clustering were also interrogated by manual gating. This revealed that the plasmacytoid DC population (CD123+ CD11c− DCs) were significantly different in frequency between the three groups ($p = 0.024$, $\chi^2 = 7.5$, $\chi^2$-test; Fig. 4c). PST infants showed a significant increase in this population relative to NA controls ($p = 0.005$, $\chi^2 = 7.8$, $\chi^2$-test).

To determine any functional differences between the clinical phenotypes, we compared cellular activation (as measured by intracellular cytokine production) after nonspecific stimulation with PMA/ionomycin. A significant difference in the frequency of TNFα+ cells across all live cells was observed between the three groups ($p = 0.015$, $\chi^2 = 8.4$, $\chi^2$-test; Fig. 4d), with PBMCs from PA infants showing more TNFα+ cells relative to NA controls ($p = 0.0079$, $\chi^2 = 7.1$, $\chi^2$-test; Fig. 4d). The median intensity of TNFα was also different across all live cells ($p = 0.0065$, $\chi^2 = 10.1$, $\chi^2$-test), with PA infants expressing higher levels of TNFα per cells than the non-allergic controls ($p = 0.017$, $\chi^2 = 5.7$, $\chi^2$-test; Supplementary Fig. 3C). Following stimulation, NK cells produced the greatest amount of TNFα (median 25.25% of NK cells across the three groups) (Supplementary Table 2). Infants with PA showed the greatest median production of TNFα for B

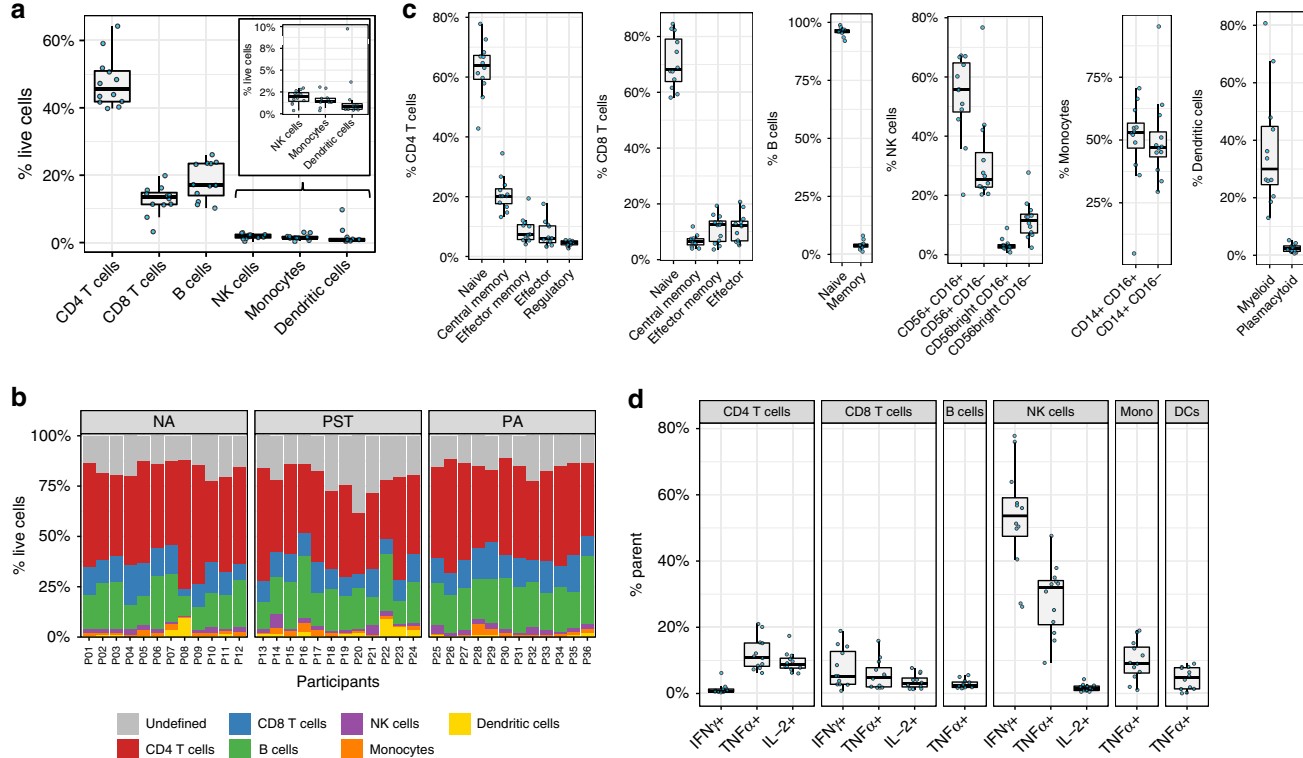

**Fig. 2 Immune cell profiling of unstimulated PBMCs from 1-year-old infants by manual gating. a** The six major immune cell populations identified in unstimulated PBMC expressed as percentage of live cells in $n = 12$ healthy 1-year-old infants. **b** Stacked bar graph representing the major immune cell populations in each individual, stratified by clinical outcome for non-allergic (NA) ($n = 12$), peanut-sensitized tolerant (PST) ($n = 12$), and peanut-allergic (PA) ($n = 12$) infants. **c** Each major immune cell population was subtyped into further populations in unstimulated PBMC from $n = 12$ healthy infants. **d** Percentage of CD4 T cells, CD8 T cells, B cells, NK cells, monocytes, and dendritic cells (DCs) producing IFNγ, TNFα, or IL-2 following 24 h stimulation of PBMC with PMA/ionomycin in $n = 12$ healthy infants. In the boxplots, the medians are shown. The "hinges" represent the first and third quartile. The whiskers are the smallest and largest values after exclusion of outliers (greater than the 75th percentile plus 1.5 times the IQR or less than 25th percentile minus 1.5 times the IQR). Source data are provided as a Source Data file.

cells, CD4 T cells, monocytes, pDCs, and mDCs, although only the pDC subset reached significance (Supplementary Table 2).

Unsupervised analysis of the PMA/ionomycin-stimulated samples revealed ten clusters of known cell populations (Supplementary Fig. 4A). We observed a significant difference in IL-2 levels within the naive CD4 T-cell cluster ($p = 0.007$, $\chi^2 = 9.8$, FDR-adjusted $p = 0.077$, $\chi^2$-test; Fig. 4e). PST infants showed higher IL-2 relative to both the PA and NA controls ($p = 0.046$, $\chi^2 = 4.0$ and $p = 0.005$, $\chi^2 = 7.9$, respectively, $\chi^2$-test). This was confirmed by manual gating, which revealed greater median IL-2 expression in naive CD4 T cells from both PA and PST infants ($p = 0.032$, $\chi^2 = 4.6$ and $p = 0.044$, $\chi^2 = 4.1$, respectively, $\chi^2$-test; Supplementary Fig. 4B). The frequency of IL-2$^+$ cells in naive CD4 T cells was also higher in PA and PST infants relative to NA controls ($p = 0.015$, $\chi^2 = 5.9$ and $p = 0.008$, $\chi^2 = 7.1$, respectively, $\chi^2$-test; Supplementary Fig. 4C).

IFNγ levels were significantly different between the three groups in the effector memory CD4 T-cell cluster expressing increased levels of HLA-DR ($p = 0.009$, $\chi^2 = 9.4$, FDR-adjusted $p = 0.077$, $\chi^2$-test; Fig. 4f). Pairwise post-hoc analyses showed significantly lower levels in PA and PST infants compared with NA controls ($p = 0.033$, $\chi^2 = 4.5$ and $p = 0.008$, $\chi^2 = 7.1$, respectively, $\chi^2$-test). These trends from the unsupervised analysis were confirmed when comparing the median IFNγ levels in manually gated effector memory CD4 HLA-DR$^+$ T cells (Supplementary Fig. 4D). When comparing our findings of the unsupervised analysis of median expression with the manually gated percent of IFNγ$^+$ effector memory CD4 HLA-DR$^+$ T cells

between the groups, the trend can be seen but is not reflected as significant changes (Supplementary Fig. 4E).

**Peanut-specific T cells in allergic but not tolerant infants.** To investigate whether the transition from sensitization to allergy involves changes in the frequency and phenotype of peanut-specific CD4 T cells, we quantified CD40L$^+$CD69$^+$ antigen-activated CD4 T cells[14] by manual gating following stimulation with pure peanut protein solution (Fig. 5a). This revealed that PA infants show an increased percentage of peanut-specific CD4 T cells (median: 0.063%, (range: 0.013–0.12%)) relative to both PST (0.026% (0.0068–0.091%)) and NA (0.038% (0.0023–0.073%)) infants following in-vitro stimulation with peanut ($p = 0.019$, F = 6.5 and $p = 0.028$, F = 5.5, respectively, F-test; Fig. 5b). Stimulation with peanut induced a median 1.79-fold increase (range: 1.17–2.18) in peanut-specific CD4 T cells in PA infants, which was significantly greater than the median 1.33-fold increase (0.39–2, $p = 0.012$, F = 7.7, F-test) in PST and the 1.22-fold increase (0.59–2.52, $p = 0.022$, F = 6.1, F-test) in NA infants (Fig. 5c). To next determine the phenotype of these peanut-specific CD4 T cells, markers of activation and memory were used to distinguish effector (CD45RA$^+$CCR7$^-$), effector memory (CD45RA$^-$CCR7$^-$), and central memory phenotypes (CD45RA$^-$CCR7$^+$). This revealed that infants with peanut allergic show more peanut-specific CD4 T cells within the CD45RA$^-$ memory phenotypes that are not observed in PST and non-allergic infants (Supplementary Fig. 5A–D).

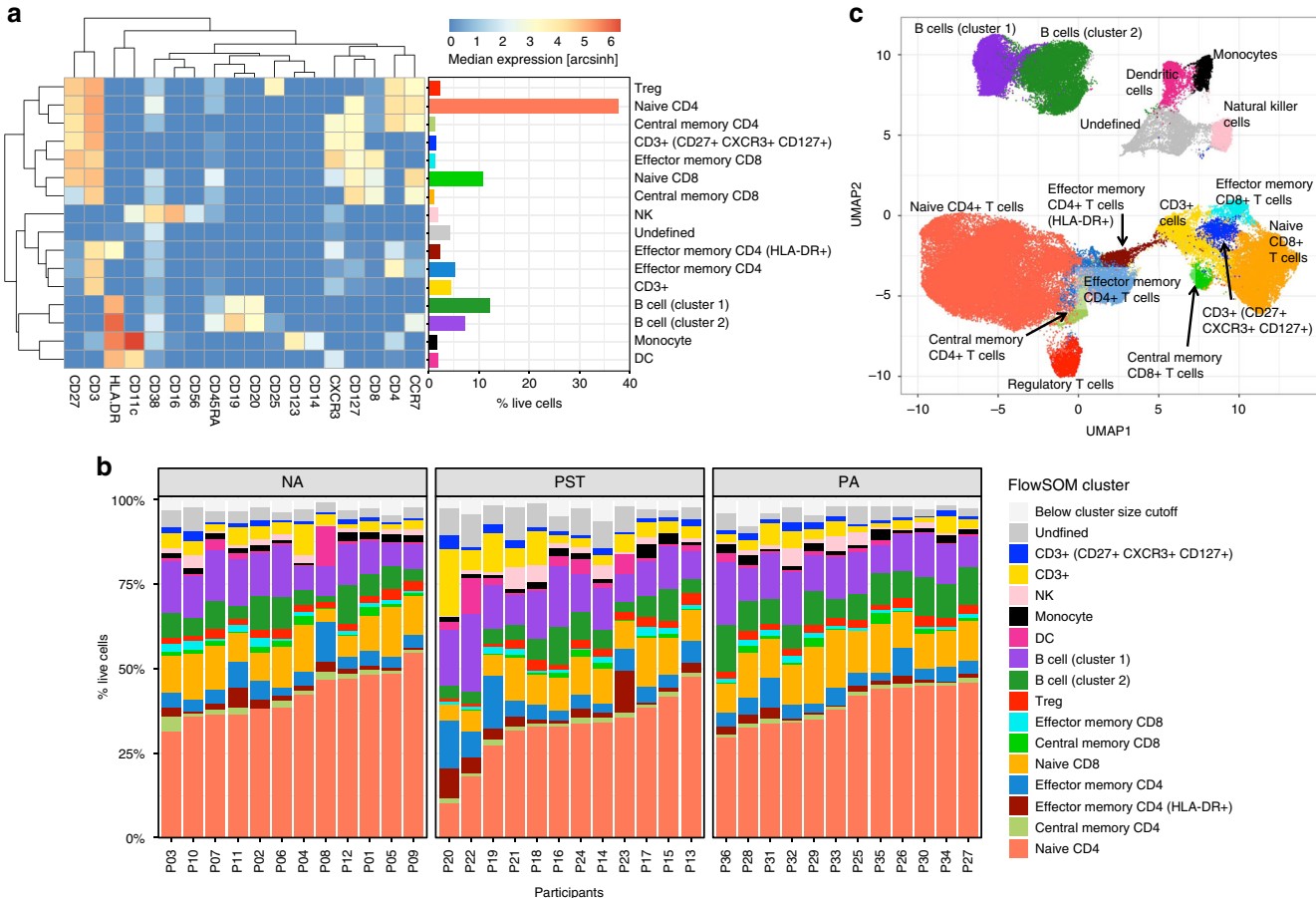

**Fig. 3 Unsupervised analysis using FlowSOM reveals distinct immune cell clusters in PBMC from 1-year-old infants. a** Clustering analysis with FlowSOM revealed 16 cell clusters, shown here as a heatmap of the median expression of 18 lineage markers along with a bar graph representing each cluster as percentage of live cells and the cell phenotype that each cluster was assigned based on lineage marker expression pattern. **b** Stacked bar graph representing all clusters identified in each individual, stratified by clinical outcome for non-allergic (NA) ($n = 12$), peanut-sensitized tolerant (PST) ($n = 12$), and peanut-allergic (PA) ($n = 12$) infants. Clusters with <1% of the analyzed cells were excluded from all other analyses and are represented here as "below cluster size cutoff". **c** Uniform Manifold Approximation and Projection (UMAP) representation of 100,008 randomly selected cells (2778 per file) with clusters from the FlowSOM analysis overlaid. The same color is used for each cluster/cell type across all plots. Source data are provided as a Source Data file.

## Discussion

This study performed high-dimensional mass cytometry-based single-cell profiling of resting and stimulated PBMCs to identify the circulating immune cell signatures associated with peanut sensitization with tolerance (PST) vs. clinical peanut allergic (PA). Unsupervised clustering and mixed-effects model analysis of cell frequency revealed that PA is associated with increased frequency of CD19$^{++}$HLADR$^{++}$ B cells compared with PST infants. In comparison, PST was associated with reduced frequency of CD45RA$^{+}$CCR7$^{+}$ naive CD4 T cells and increased frequency of plasmacytoid DCs compared with healthy infants. Functional analysis of intracellular cytokine production following PMA/ionomycin stimulation revealed increased global expression of TNFα across all single cells in PA infants and increased IL-2 expression in the naive CD4 T-cell cluster in PST infants. In addition, investigation of the peanut-specific T-cell response by expression of CD40L and CD69 on CD4 T cells following peanut stimulation revealed that PA, but not PST, is characterized by increases in peanut-activated CD4 T cells that display a memory phenotype.

These findings highlight the utility and power of high-dimensional mass cytometry, in combination with unsupervised and manual analysis techniques, to interrogate the cellular interactions that could explain the development of allergic sensitization with tolerance vs. clinical food allergy in the first year of life. Importantly, analysis by mass cytometry revealed cell frequencies comparable to that obtained previously with flow cytometry in the same age-matched HealthNuts cohort[8–10]. In the current study, the key features associated with PA were increased frequency of a unique subset of B cells (CD19$^{++}$HLADR$^{++}$) and overproduction of TNFα following a nonspecific stimulation. We and others have previously shown an increased production of inflammatory cytokines, including TNFα, from myeloid cells in food-allergic patients[9,15]. In the current study, by far the greatest producers of TNFα following stimulation were NK cells, followed by CD4 T cells and monocytes, with a small proportion of DCs, CD8 T cells, and B cells also contributing to this global inflammatory response. The PA group showed the greatest mean production of TNFα across NK cell, CD4 T cell, monocyte, DC, and B-cell populations. Together, these findings confirm that highly complex cellular inflammatory responses play an important role in the development of food allergy. Interestingly, a recent study investigating the phenotype of peripheral B cells in systemic lupus erythematosus (SLE) identified a similar increase in CD19$^{++}$HLADR$^{++}$ cells in SLE patients relative to healthy controls[16]. The authors showed this

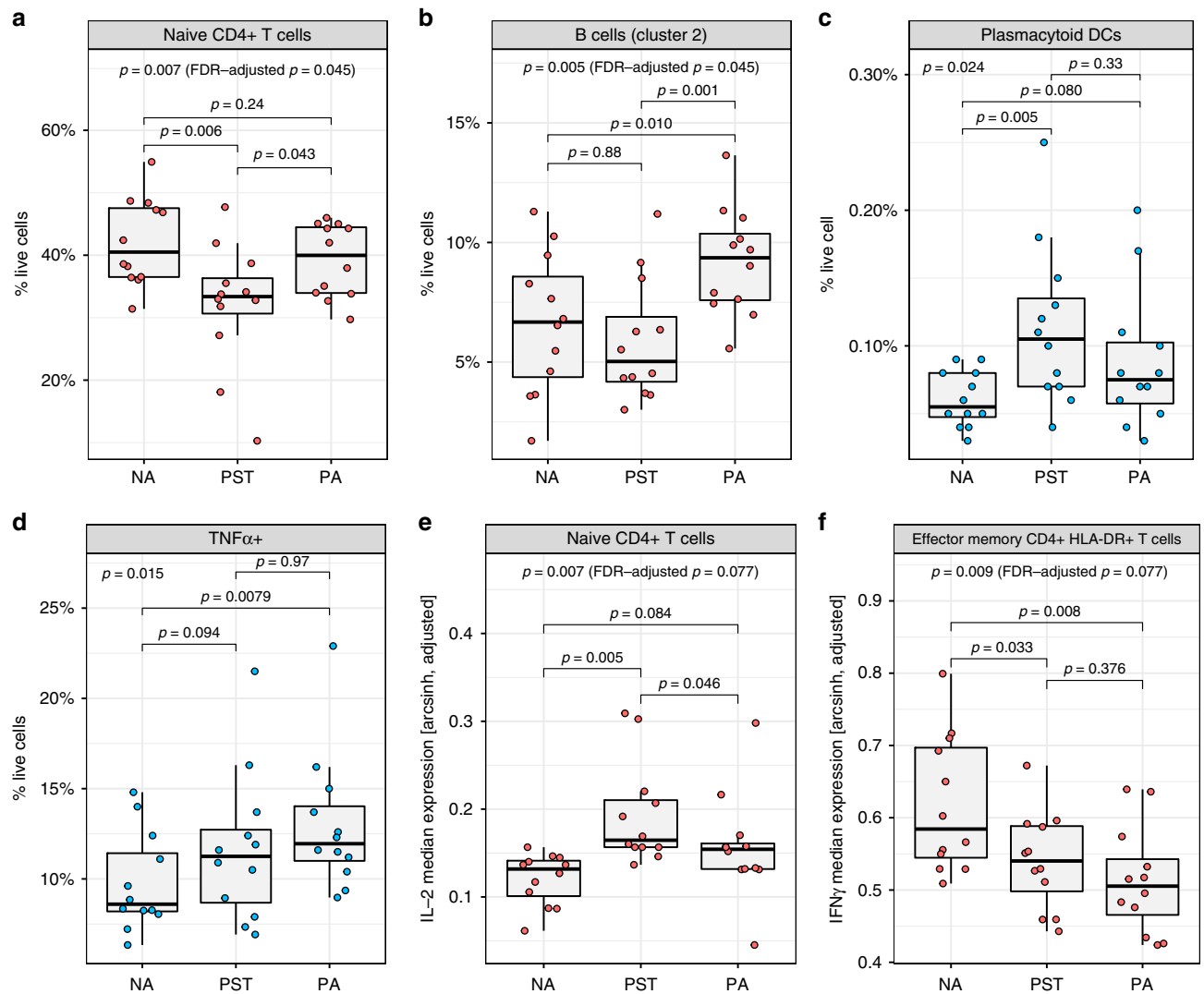

**Fig. 4 Unsupervised and manual gating analysis of resting and PMA/ionomycin-stimulated PBMCs reveals differences in multiple immune signatures between non-allergic (NA), peanut-sensitized tolerant (PST), and peanut-allergic (PA) infants.** Frequency (as % of live cells) of (**a**) naive CD4 T-cell cluster and (**b**) B-cell cluster 2 (CD19++HLADR++) identified by unsupervised analysis in unstimulated PBMCs from NA, PST, and PA infants. **c**, **d** Frequency (as % of live cells) of resting plasmacytoid DCs (CD123+CD11c−) and TNF-α+ cells as identified by manual gating from NA, PST, and PA infants. Median expression of (**e**) IL-2 in the naive CD4 T-cell cluster and (**f**) IFNγ in effector memory HLADR + CD4 T-cell cluster as identified by unsupervised analysis in NA, PST, and PA infants. P-values by $\chi^2$-tests in mixed-effects model analyses, between the three groups and post-hoc pairwise. Median expression values were adjusted for batch before plotting. Blue dots represent values determined by manual gating, whereas red dots represent values derived from the unsupervised analysis. In the boxplots, the medians are shown. The "hinges" represent the first and third quartile. The whiskers are the smallest and largest values after exclusion of outliers (greater than the 75th percentile plus 1.5 times the IQR or less than 25th percentile minus 1.5 times the IQR). Source data are provided as a Source Data file.

subset of cells produced more immunoglobulin relative to the CD19+ population and showed an upregulation of genes involved in B-cell activation and differentiation. Whether this activated B-cell subset is responsible for the increased IgE observed in PA needs to be further investigated in in-vitro studies.

Both a reduced frequency of naive CD4 T cells and increased expression of IL-2 in this cell subset following stimulation were signatures uniquely associated with PST. IL-2 has recently been described to be critical in the maintenance of immune tolerance (comprehensively reviewed in ref. [17]). Clinical research has shown that IL-2 therapy is effective in promoting tolerance in autoimmune and inflammatory conditions, primarily by enhancing the development and maintenance of regulatory T cells[18]. In this study, we additionally report that an increase in plasmacytoid DCs, also known for their tolerogenic immune properties against

innocuous antigens[19,20], to be unique to infants with PST. These cells may play a central role in averting clinical allergy in the presence of allergen-sIgE.

Few studies have investigated the immune response of infants who are sensitized but clinically tolerant to peanut. Previous work investigating the plasma cytokine profiles of 119 infants with sensitized tolerance or clinical allergy to food (peanut or egg) showed that IL-10 and IL-6 were significantly higher in those sensitized compared with those who were allergic, and that IL-4 and IL-13 were higher in sensitized relative to non-sensitized infants[21]. Basophil-activation tests in PA children showed upregulation of basophil-activation markers CD63 and CD203c following peanut stimulation; however, there was no response to peanut in those who were peanut-sensitized but tolerant[22]. We also examined the peanut-specific response in

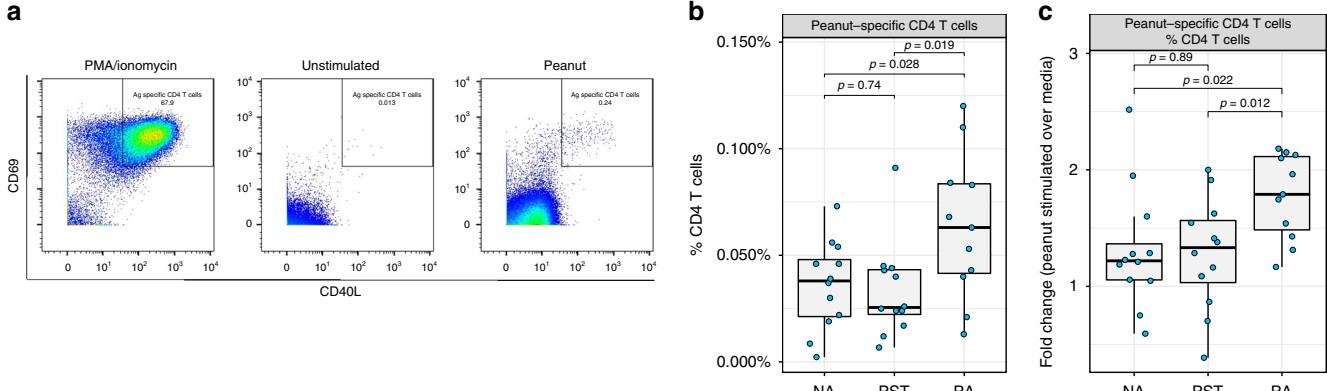

**Fig. 5 Analysis of manually gated peanut-specific CD4 T cells shows significant differences between peanut-sensitized tolerant (PST) and peanut-allergic (PA) infants. a** Peanut-specific CD4 T cells were identified in peanut-stimulated cultures based on upregulation and co-expression of T-cell activation markers CD40L and CD69. The positive control, PMA/ionomycin stimulation, was used to set the gates to identify these peanut-activated T cells in all groups. Peanut-specific CD4 T cells were identified in PA infants ($n = 11$), PST infants ($n = 12$), and NA infants ($n = 12$), and expressed as (**b**) proportion of CD4 T cells and (**c**) fold change after peanut stimulation over unstimulated (media) conditions. $P$-values by F-tests in linear models. In the boxplots, the medians are shown. The "hinges" represent the first and third quartile. The whiskers are the smallest and largest values after exclusion of outliers (greater than the 75th percentile plus 1.5 times the IQR, or less than 25th percentile minus 1.5 times the IQR). Source data are provided as a Source Data file.

this cohort by analyzing upregulation of CD40L and CD69 as markers of antigen-stimulated CD4 T cells following peanut stimulation, as previously described[14,23,24]. We showed an increase in peanut-specific effector CD4 T cells following peanut stimulation in PA infants; however, no response to peanut in those who were PST and NA. Previous work investigating peanut-specific T-cell responses have been hindered by technical difficulties in identifying these rare cells in the circulation. Tetramer-based technology provides specific and sensitive measures of allergen specificity; however, they require T-cell expansion cultures and knowledge of HLA (human leukocyte antigen) status[25–27]. Evaluating peanut specificity via the CD154 and CD69 upregulation assay reported in our study, both markers of T-cell receptor activation, allows improved detection of these rare cells after a short-term culture with endotoxin-free pure peanut solution.

We acknowledge this study has several limitations. Although comprehensive for the well-characterized lymphoid and myeloid markers, a limitation of our antibody panel is that it does not include surface markers for innate-like and unconventional T-cell populations, or subsets of immunoglobulin switched/non-switched B cells, which may be relevant for the the development of an allergic immune response[28,29]. It should also be considered when interpreting our findings that, as blood collection was performed following clinical testing, a positive peanut OFC could influence some of the immune parameters investigated in this study. As such, the results presented represent a snapshot of the immune response following in vivo allergen exposure. However, we have previously reported no differences in cellular activation or plasma cytokine production in food-allergic infants who had a blood sample taken on a non-OFC day vs. an active OFC day[9,21]. It will be interesting to determine whether these signatures can be identified prior to allergen challenge and we acknowledge these are important areas for future investigation into the development of peanut sensitization and allergy in the first year of life.

This study uses mass cytometry to provide a detailed characterization of the circulating immune cell profiles that are associated with challenge-proven food allergy and asymptomatic food sensitization in the first year of life. The results provide a framework for future investigation into the roles of these unique immune signatures in predicting the transition from sensitization to allergy.

## Methods

**Oral food challenges and serum sIgE.** OFC to peanut were performed in all subjects according to standardized protocols[30]. Briefly, food challenge involved gradually increasing doses on day 1 in the hospital and continued ingestion of the maximum day 1 dose (1 teaspoon of peanut butter) on days 2 through 7. HealthNuts-standardized cessation criteria for a positive OFC result were any of the following occurring within 2 h of ingestion: three or more concurrent non-contact urticaria persisting for at least 5 min, perioral or periorbital angioedema, vomiting, and evidence of circulatory or respiratory compromise[30]. Serum sIgE was measured using the ImmunoCAP System FEIA (Phadia AB). Values at the lower detection limit of 0.01 kUA/L were set to 0.005 kUA/L.

**Cell culture and preparation of cells for mass cytometry.** Blood was collected at clinic appointments 2 h following peanut OFC and PBMCs were isolated from whole blood by density gradient and cryopreserved as detailed previously[11]. Briefly, up to 7 mL of blood was collected from each infant into a sodium heparin tube (Sarstedt) and plasma removed by centrifugation within 1 h of collection at $700 \times g$ for 10 min at room temperature. A 1:1 ratio of RPMI media was added to cells before layering onto 5.0 mL of Ficoll-Paque solution and brake-free centrifugation at $400 \times g$ for 30 minutes. Mononuclear cells at the interface of media and Ficoll-Paque solution were aspirated and washed twice in RPMI containing 2% heat-inactivated fetal calf serum (FCS) by centrifugation at $500 \times g$ for 7 min. PBMCs were cryopreserved in liquid nitrogen at $10 \times 10^6$/ml in RPMI with 15% dimethyl sulfoxide in FCS.

For cell culture, PBMCs were thawed in 10 mL cell culture media (RPMI supplemented with 10% heat-inactivated FCS and penicillin streptomycin) with 25 U/mL benzonase at 37 °C. PBMCs were centrifuged at $300 \times g$ for 10 min and washed twice in culture media before viability count using the NucleoCounter NC-200. Mean viability after thawing was 90.5%. Cells were resuspended at $2 \times 10^6$/mL in cell culture media for overnight rest in a T25 flask at 37 °C, 5% $CO_2$. Following overnight rest, cells were then resuspended at $3 \times 10^6$/200 μL and cultured in U-bottom 96-well plates with ether (i) media alone, (ii) 200 μg/ml of endotoxin cleaned pure peanut protein solution (Greer: XPF171D3A2.5: Ara h 1 content: 71.03 μg/mL, Ara h 2 content: 78.43 μg/mL) for 24 h or (iii) 20 ng/mL PMA/1 μg/mL ionomycin combined solution for the final 4 h. PMA/ionomycin was chosen as a nonspecific cell stimulus and as a positive control in our assay to ensure cells were responsive to stimulation. To inhibit extracellular cytokine transport, Brefeldin-A was added to all wells after 20 h. Following cell culture, PBMCs were centrifuged at $300 \times g$ for 7 min, resuspended in 200 μl-filtered CyFACS buffer (0.1% bovine serum albumin, 0.1% sodium azide, 2 mM EDTA in PBS) and transferred to V-bottom 96-well plates for staining.

All of the following cell staining steps prior to barcoding were performed in V-bottom 96-well plates, with wash steps in 200 μl CyFACS buffer and centrifugation at $300 \times g$ for 7 min. PBMCs were resuspended in 70 μl of surface antibody cocktail (Supplementary Table 1) and incubated for 30 min at room temperature. Cells were then washed three times and resuspended in 100 μl of live/dead 115-DOTA

maleimide (stock 5 mg/ml, diluted 1:3000) for 15 min at room temperature. Cells were then washed a further three times prior to transfer into polypropylene fluorescence-activated cell sorting tubes and barcoding using the Cell-ID 20-Plex Pd Barcoding Kit (Fluidigm) according to manufacturer's instructions. PBMCs were then resuspended in 100 µl of 2% paraformaldehyde (PFA) in CyPBS (filtered PBS) and incubated overnight at 4 °C. The next day, cells were resuspended in 2 ml CyFACS buffer and centrifuged at $600 \times g$ for 5 min at 4 °C. Following cell count, an equal number of cells from each infant were pooled into a single 15 ml tube and centrifuged at $600 \times g$ for 5 min at 4 °C. For permeabilization, cells were resuspended in 2 ml of permeabilization buffer (EBioscience) and centrifuged at $600 \times g$ for 5 min at 4 °C. Following a second wash in 2 ml permeabilization buffer, pooled cells were resuspended in 100 µl of intracellular antibody cocktail (Supplementary Table 1) and incubated for 30 min at room temperature. Cells were then washed once in 2 ml of permeabilization buffer, followed by two washes in 2 mL CyFACS buffer. For every sample within the pooled tube, 100 µl of Ir-Interchelator (1:2000, diluted in 2% PFA in CyPBS) was added and incubated overnight at 4 °C. On the day of mass cytometry acquisition, cells were washed twice in CyFACS buffer, followed by one wash in CyPBS and two further washes in milliQ water. All wash volumes were in 2 ml and centrifugation was at $600 \times g$ for 5 min at 4 °C. Samples were acquired on the mass cytometer (Helios, Fluidigm) after standard instrument setup procedures.

**Mass cytometry data analysis**. Following normalization and de-barcoding, FCS files underwent standard pre-processing to remove debris, doublets and to enrich for live cells (Supplementary Fig. 6). Live, single cells were analyzed by manual gating and unsupervised computational methods in parallel.

For manual gating, major immune cell populations were identified based on the gating strategy outlined in Supplementary Fig. 7 using FlowJo V10.6. Briefly, CD3 and CD19 were first used to determine pan T and B cells, respectively. Within the B-cell population, naive and memory B cells were identified based on CD27 expression. Within the T-cell population, CD4 and CD8 T cells were identified. For CD8 T cells, naive, effector, central memory, and effector memory subpopulations were identified based on CD45RA and CCR7 expression. For CD4 T cells, Tregs were determined based on $CD25^+CD127^{low}$ expression and a "not Treg" gate was created to then identify CD4 naive, effector, central memory, and effector memory subpopulations based on CD45RA and CCR7 expression. NK cells were identified from within the $CD3^-CD19^-$ population based on CD56 expression. Subpopulations of NK cells were also determined based on the brightness of CD56 and expression of CD16. Finally, $CD3^-CD19^-CD56^-$ cells were then separated into monocytes, based on CD14 expression (subpopulations of monocytes also identified based on CD16 expression) and DCs based on $CD14^-HLADR^+$ expression, where DC subsets were determined by CD11c (myeloid) and CD123 (plasmacytoid) expression. CD49b and LAG3 were excluded from manual and computational analysis due to limited expression across all samples. For cytokine production following 24 h stimulation with PMA/ionomycin, IL-2, TNFα, and IFNγ expression were analyzed in all cell populations (Supplementary Fig. 8). IL-10, IL-4, and IL-17A were excluded from manual and computational analysis due to limited expression across all samples.

For quantification of antigen-specific CD4 T cells following peanut stimulation, $CD40L^+CD69^+$ cells were identified within the CD4 T-cell population by manual gating[14]. For one of the PA infants, due to limited cell number, no peanut stimulation was performed, reducing the size of that group for this analysis to $n = 11$. To analyze the impact of peanut stimulation on the frequency of allergen-specific cells within CD4 T cells between the three groups, per infant, the fold change of frequency of these cells after peanut stimulated over the frequency when unstimulated was calculated. Pairwise linear models with F-test were used to compare the determined frequencies or fold changes between the three groups. $P < 0.05$ was considered significant.

Unsupervised computational analysis was performed separately for unstimulated and PMA/ionomycin-stimulated samples. In each case, 50,000 cells of the pre-gated live, single cells were randomly selected and values were arcsinh (inverse hyperbolic sine) transformed with a cofactor of 5[31]. As introduced previously as a strategy to analyze mass cytometry data[32], we separated the markers into lineage markers (CD3, CD4, CD8, CCR7, CD45RA, CD25, CD127, CD27, CXCR3, CD19, CD20, CD38, CD56, CD14, CD16, HLA-DR, CD11c, and CD123) and functional markers (IFNγ, TNFα, and IL-2). For the nonspecifically stimulated samples CD127, CXCR3 and CD27 were excluded due to batch effects. To adjust for between-sample and batch variation, a subset of the lineage markers (all but CD45RA and CD38 for unstimulated and all for the PMA/ionomycin-stimulated samples) were subjected to the landmark alignment procedure[33], using the warpSet R function in the flowStats package (version 3.40.1). The warping functions were determined using all live, single cells per file and not just the randomly selected cells. Subsequently, unsupervised clustering was performed on the expression values of the lineage markers using the FlowSOM algorithm[13] (R package FlowSOM, version 1.14.1), which uses a self-organizing map followed by hierarchical consensus meta-clustering to detect cell populations. Default parameters and a predetermined number of 25 clusters were used. Clusters with <1% of the analyzed cells were excluded.

The median levels of the lineage markers across all cells per cluster were visualized in a heatmap for the unstimulated and PMA/ionomycin-stimulated samples (R package pheatmap, version 1.0.12[34]). The type of cells within a cluster was based on the levels of expression of specific lineage markers. For the unstimulated samples, we applied the non-linear dimensionality reduction technique UMAP to the lineage marker levels of a set of randomly selected 100,008 cells (2778 cells per file) using the R package UMAP (version 0.2.0.0, default parameters except min_dist = 0.25) for visualization of the high-dimensional data[35]. The cells were colored according to their FlowSOM cluster membership. Clusters for which no known cell type could be identified based on surface marker expression were included in the heatmap and UMAP but were excluded from the subsequent analysis.

For the unstimulated samples, the percentage of the randomly selected cells of each file in each cluster was determined and the values were compared between the three groups per cluster using a linear mixed-effects model with batch (five batches) as random effect. $P$-values were determined by $\chi^2$-test (2 degrees of freedom (DF)) and adjusted for multiple comparisons using the Benjamini and Hochberg approach to control the FDR[36]. In this analysis, 13 tests were performed and FDR-adjusted $p < 0.1$ were considered significant. For each of the clusters showing a significant difference between the three groups, a descriptive post-hoc analysis by a series of additional mixed-effects models and $\chi^2$-tests (1 DF) each comparing two groups were performed.

After clustering using FlowSOM, for the PMA/ionomycin-stimulated samples, median levels of the functional markers per cluster per file were compared between the three groups in the mixed-effects models. $P$-values by $\chi^2$-test were FDR-adjusted and features significantly different between the three groups (FDR-adjusted $p < 0.1$, total number of tests: 25) were analyzed in pairwise post-hoc mixed-effects models. Median levels were plotted after adjustment for batch in linear models.

Manual gating was used to confirm detected significant changes from the unsupervised analysis as well, to study rare and specific cell types not detected by the unsupervised analysis. $P$-values were determined by $\chi^2$-tests in mixed-effects models as in the unsupervised analysis.

In all presented boxplots, the medians are shown. The "hinges" represent the first and third quartile. The whiskers are the smallest and largest values after exclusion of outliers (greater than the 75th percentile plus 1.5 times the interquartile range (IQR), or less than 25th percentile minus 1.5 times the IQR). When data points for each individual were overlaid on the boxplots, blue dots represent values determined by manual gating, whereas red dots represent values derived from the unsupervised analysis using FlowSOM. The statistical analysis was performed in R (version 3.5.2) and plotting was done using the ggplot2 R package (version 3.1.0)[37]. All statistical tests were performed two-sided.

**Ethics**. Approval to conduct the HealthNuts study was obtained from the Victorian State Government Office for Children (reference number CDF/07/492), the Victorian State Government Department of Human Services (reference number 10/07), and the Royal Children's Hospital Human Research Ethics Committee (reference number 27047). Informed consent was obtained from parents or guardians of all participants.

**Reporting summary**. Further information on research design is available in the Nature Research Reporting Summary linked to this article.

## Data availability
The source data underlying all figures and supplementary figures are provided as a Source Data file.

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

## Acknowledgements

We thank the children and parents who participated in the HealthNuts study. We also thank current and past staff for recruiting and maintaining the cohort, and processing the data and biospecimens. We thank Matthew Kirkey for assistance with CyTOF experiments. This work was supported by funding from the NIH (5R01AI140134, 2U19AI104209, and 5U01AI140498), EAT (End Allergies Together), FARE (Food Allergy Research & Education), the Sean N. Parker Center for Allergy and Asthma Research at Stanford University, and the Kim and Ping Li endowment. The HealthNuts study is supported by funding from the National Health and Medical Research Council of Australia (NHMRC), the Ilhan Food Allergy Foundation, AnaplyaxiStop, The Charles and Sylvia Viertel Medical Research Foundation, and the Victorian Government's Operational Infrastructure Support Program. M.R.N. is supported by a Melbourne Children's Lifecourse Postdoctoral Fellowship.

## Author contributions

M.R.N. designed and performed the experiments, analyzed the data, and wrote the manuscript. S.A. analyzed the data and wrote the manuscript. M.M., D.D., and S.L. contributed to experimental design and assisted with experiments. T.D. collected and processed the biospecimens. K.P., M.T., R.P., R.S., and J.K. are HealthNuts investigators and contributed to study conceptualization. K.C.N. supervised the work, provided funding, and edited the manuscript. All authors edited and approved the final manuscript.

## Competing interests

K.C.N. reports grants and other from NIAID other from Novartis, personal fees and other from Regeneron, grants and other from FARE, grants from EAT, other from Sanofi, other from Astellas, other from Nestle, other from BeforeBrands, other from Alladapt, other from ForTra, other from Genentech, other from AImmune Therapeutics, other from DBV Technologies, personal fees from Astrazeneca, personal fees from ImmuneWorks, personal fees from Cour Pharmaceuticals, grants from Allergenis, grants from Ukko Pharma, other from AnaptysBio, other from Adare Pharmaceuticals, other from Stallergenes-Greer, other from NHLBI, other from NIEHS, other from EPA, other from WAO Center of Excellence, other from Iggenix, other from Probio, other from Vedanta, other from Centecor, other from Seed, from Immune Tolerance Network, from NIH, outside the submitted work. In addition, K.C.N. has a patent Inhibition of Allergic Reaction to Peanut Allergen using an IL-33 Inhibitor pending, a patent Special Oral Formula for Decreasing Food Allergy Risk and Treatment for Food Allergy pending, a patent Basophil Activation Based Diagnostic Allergy Test pending, a patent Granulocyte-based methods for detecting and monitoring immune system disorders pending, a patent Methods and Assays for Detecting and Quantifying Pure Subpopulations of White Blood Cells in Immune System Disorders pending, a patent Mixed Allergen Compositions and Methods for Using the Same pending, and a patent Microfluidic Device and Diagnostic Methods for Allergy Testing Based on Detection of Basophil Activation pending.
