## [Peer Review File · Nature Communications]

Reviewers' comments:

Reviewer #1, expert in mass cytometry (Remarks to the Author):

Thank you for the opportunity to review the manuscript by Neeland et al.: "Mass cytometry reveals multiparametric immune signatures that govern allergy over tolerance in peanut sensitized infants". The authors use a mass cytometry approach to compare the distribution of peripheral immune cells and select intracellular cytokine responses in infants with clinical allergy to peanut (PA), sensitized but tolerant infants and healthy non-sensitized infants. Immune signatures that differentiate the three groups include: increased CD19+HLADR+B cell fq and increased total TNFa expressing cells in PA compared to controls, while PST patients had lower CD4+T naïve frequency, higher pDC frequency, and higher IL-2 producing CD4+T cell frequency than healthy controls. In addition, PA patients had higher peanut sensitive CD4+T cells than PST or controls. This is an interesting and well written manuscript that addresses the important and clinically relevant issue of identifying a biomarker distinguishing PST from PA patients early in life. The application of the high dimensional immunoassay is technically sound and the use of unsupervised approaches for immune feature identification followed by validation with manual gating is commendable. The manuscript could benefit from addressing the following comments.

1- The high dimensional analysis of clinical dataset produced interesting features in the baseline as well as stimulated samples. Readers would benefit from a more cohesive, unifying interpretation of the results that attempts at relating functional and cell distribution findings. Specifically, it is unclear how the increased TNFa production observed in PBMCs from PA patients relates to changes in specific cell frequencies. The authors refer to previous work suggesting that myeloid cells may be the predominant TNFa producers. It would be surprising that this the case in the current dataset given that myeloid cells represent such small proportion of total PBMC (especially after PMA/ionomycin treatment). Similarly, the interpretation of increased IL-2 production in PST as a marker of tolerance is interesting. In this dataset, does IL-2 production correlate with increased Treg frequencies and/or function?

2- The statistical approach comparing the three groups is robust and appropriate. However, with the limited study size (12 patient per group), comparing three groups may be "too stringent" to allow identification of the most clinically relevant immune features (PA vs PST group). A less stringent analysis comparing the PA vs PST group only may reveal additional and clinically important immune differences that could help in interpreting the major findings.

3- A justification for the choice of PMA/ionomycin as a stimulation is warranted, given that a more physiological stimulation is used in the manuscript. Similarly, the choice of the 6 intracellular cytokines included in the panel needs to be discussed.

4- Differences in TNFa production across all cell types could be the result of non-specific differences between groups, including sample collection and processing, instrument-related batch effect etc... A description of the precautions taken to mitigate such batch effects when possible is warranted.

Brice Gaudilliere, MD, PhD

Reviewer #2, expert in food allergy (Remarks to the Author):

The topic of this manuscript, food allergy, belongs to the most dangerous epidemics around the world. The mechanisms by which infants get sensitized, and then develop towards sensitized but tolerant PST, or true food allergic PA, are largely not understood. Peanut is a paradigm of importance in the industrialized world with a lifelong risk of severe adverse events. The understanding how the tolerant state is transferred into the allergic is an unmet need. The presented work is novel and original.

From a well characterized clinical sample, the HealthNuts cohort, 3 groups (each n=12) were recruited: In 36 infants being PST, PA or healthy (NA), carefully clinically characterized according to IgE values, SPT reactivity and reactivity upon oral challenge to peanut. From these cohorts, blood cells were isolated and examined by high dimensional mass cytometry-based single cell profiling. The frequency of cell species clusters was typed by manual gating, as well as by unsupervised analysis. Overall, the authors aim to explain the development of allergic sensitization with tolerance vs. clinical food allergy in the first year of life, while the results are a collection of observations in cellular phenotypes. The results do not allow a dynamic picture of how allergy develops and do not allow correlations with the elevated IgE in PA children, as the authors state themselves in line 235.

Significant differences were found between T cells, B cells and plasmacytoid DCs:

A) in PA children reduced frequency of the naïve CD4 T cell cluster and

B) in PA children increases in peanut-specific memory CD4 T cells, CD45RA- memory phenotypes

C) in PST in PST kids higher frequency of B cell cluster 2 (CD19++HLADR++ B cell) (compared to both groups PA or NA),

D) in PST in PST kids higher frequency of plasmacytoid DCs (CD123+ CD11c- DCs) compared to NA.

Possibly the most important finding is that PST infants showed higher IL-2 levels with a functional role in tolerance, while for unknown reasons the cellular sources of IL2, IL-2+ cells were again comparable in naïve CD4 T cells in PA and PST infants.

On the other hand, PA showed higher potency for TNFalpha production by PBMC. This is really interesting as TNFalpha is uttermost important in anaphylaxis and prompts the question, at which time point the PBMCs were actually taken and cryo-conserved after the peanut challenges? Needs to be clarified in M&M. Under any circumstances cells were post challenge. This is pivotal as the observed signature in PA children may be a signature of the previous anaphylactic event during oral allergen challenge, instead of a clinical biomarker for the PA phenotype. Is it the chicken or the egg? Similarly, are the increases in peanut-activated CD4 T cells that display a memory phenotype in the PA group a memory of the recent allergen challenge?

There is only one way to answer this: Defining a cellular signature before oral allergen challenge. To strengthen the conclusions, can the authors identify children at risk by this phenotype before doing an oral challenge?

Minor:

Apart from the IgE and cellular data, the determination of IgG4 could be very interesting and potentially discriminate the groups.

It is remarkable that the sensitivity of the cellular tests did not allow to determine IL4 and IL10, as these are really interesting biomarkers, but were they determined in the supernatants from stimulated samples?

Finally, stimulation was done with pure peanut protein solution. In the light of the current discussion, the analysis of the extract and its compounds is missing as supplementary data. Which of the major allergens were contained, roasted or not, lipid content?

Reviewer #3, expert in pediatric food allergy (Remarks to the Author):

General Comments: In the study by Neeland and colleagues, the investigators utilized high dimensional mass cytometry-based immune profiling of resting and stimulated peripheral blood mononuclear cells (PBMCs) to define the circulating immune cell signatures associated with peanut-sensitized but tolerant infants (PSTs) vs. peanut-allergic infants (PA) vs. non-allergic healthy controls (NA) in the first year of life.

Key features associated with PA in this cohort were the increased frequency of a unique subset of B

cells (CD19++HLADR++) and over-production of TNF α following non-specific stimulation. Infants with PST were found to have a reduced frequency of (but hyper-responsive) naïve CD4 T cells and an increased frequency of plasmacytoid dendritic cells. Peanut-specific T cell responses were also evaluated and showed that infants with peanut allergy, but not PST, had higher proportions of memory peanut-specific CD4 T cells. The manuscript is well-written, and the methods clearly described. Results showing the difference in cell populations might be easier appreciated in tabular form instead of writing them out as currently in the manuscript. While technically a very impressive study, given the small numbers of subjects studied and the variability among patient profiles seen, it is not clear how well this can be used to distinguish individual patients who are peanut allergic from those who are only sensitized.

Specific Comments:

Lns 70-72: What criteria were used to select the various subsets of these cohorts from the HealthNuts population? Given the small numbers, this could significantly affect the outcome.

Lns 80-81: Is this sentence necessary? By definition the PA and PST subsets had to have evidence of IgE and the normal controls could not have positive skin tests or peanut-specific IgE.

Lns 85-137: The percentages of different cell types in this section would be easier to compare if presented in tabular form.

Lns 294-295: Was the proportion of live cell types following thawing the same as the proportion of fresh PBMC cell types, especially APCs?

Lns 388-389: Given the number of comparisons made, is an FDR set at $p < 0.1$ adequate?

Dear Dr. Bondar,

We thank you and the reviewers for the time taken to consider our manuscript and the thoughtful comments. Please find attached the revision of our manuscript.

Please note the point-by-point responses to reviewers' comments:

Reviewer #1, expert in mass cytometry (Remarks to the Author):

Thank you for the opportunity to review the manuscript by Neeland et al.: "Mass cytometry reveals multiparametric immune signatures that govern allergy over tolerance in peanut sensitized infants". The authors use a mass cytometry approach to compare the distribution of peripheral immune cells and select intracellular cytokine responses in infants with clinical allergy to peanut (PA), sensitized but tolerant infants and healthy non-sensitized infants. Immune signatures that differentiate the three groups include: increased CD19+HLADR+B cell fq and increased total TNF α expressing cells in PA compared to controls, while PST patients had lower CD4+T naïve frequency, higher pDC frequency, and higher IL-2 producing CD4+T cell frequency than healthy controls. In addition, PA patients had higher peanut sensitive CD4+T cells than PST or controls. This is an interesting and well written manuscript that addresses the important and clinically relevant issue of identifying a biomarker distinguishing PST from PA patients early in life. The application of the high dimensional immunoassay is technically sound and the use of unsupervised approaches for immune feature identification followed by validation with manual gating is commendable. The manuscript could benefit from addressing the following comments.

1- The high dimensional analysis of clinical dataset produced interesting features in the baseline as well as stimulated samples. Readers would benefit from a more cohesive, unifying interpretation of the results that attempts at relating functional and cell distribution findings. Specifically, it is unclear how the increased TNF α production observed in PBMCs from PA patients relates to changes in specific cell frequencies. The authors refer to previous work suggesting that myeloid cells may be the predominant TNF α producers. It would be surprising that this the case in the current dataset given that myeloid cells represent such small proportion of total PBMC (especially after PMA/ionomycin treatment).

RESPONSE: We thank the reviewer for these helpful comments. We have added the suggested analyses (Table S2), results (line 154-157) and discussion sections (lines 217-223) to provide more discussion of the relationship between the functional responses and resting cell distributions. Following PMA/ionomycin stimulation, TNF α was produced by various cell types (now listed in Table S2). The PA group showed the greatest mean percentage of TNF α producing cells for B cells, CD4 T cells, NK cells, monocytes, pDC and mDC when comparing the three groups, however this only reached statistical significance for the pDC population when comparing PA vs NA ($p=0.0058$) and PA vs PST ($p=0.037$) (Table S2). For CD8 T cells, PST and NA infants showed greater mean percentage of TNF α than the PA group, although not significant. It is likely that the observed increase in TNF α in the PA group is a result of a greater production of TNF α within multiple cell types, in combination with a change (even if not significant) in cell frequency. We have added this to the discussion (lines 217-223).

Similarly, the interpretation of increased IL-2 production in PST as a marker of tolerance is interesting. In this dataset, does IL-2 production correlate with increased Treg frequencies and/or function?

RESPONSE: Thank you for this interesting question. IL-2 production, as proportion of live cells or within the naïve CD4 T cell cluster, did not correlate with Treg frequency or function.

2- The statistical approach comparing the three groups is robust and appropriate. However, with the limited study size (12 patient per group), comparing three groups may be “too stringent” to allow identification of the most clinically relevant immune features (PA vs PST group). A less stringent analysis comparing the PA vs PST group only may reveal additional and clinically important immune differences that could help in interpreting the major findings.

RESPONSE: Thank you for this comment. We performed the analysis between the PA and PST groups only. The results confirmed what we found in our comparison between the three groups and no additional differences were determined.

3- A justification for the choice of PMA/ionomycin as a stimulation is warranted, given that a more physiological stimulation is used in the manuscript. Similarly, the choice of the 6 intracellular cytokines included in the panel needs to be discussed.

RESPONSE: PMA/ionomycin was used as a non-specific cell stimulus and as a positive control in our assay to ensure cells were responsive to stimulation. Peanut stimulation was used as the allergen-specific stimulation. This has been added to the methods lines 297-299.

4- Differences in TNF α production across all cell types could be the result of non-specific differences between groups, including sample collection and processing, instrument-related batch effect etc... A description of the precautions taken to mitigate such batch effects when possible is warranted.

RESPONSE: All samples were collected and processed (for PBMC isolation as well as mass cytometry) according to standardised protocols. When preparing cells for mass cytometry, sample groups (PA, PST and NA) were randomised to minimise batch effects when performing the assay on different days. From a computational approach, batch effects (where a batch is defined as being one CyTOF run), were addressed on two levels: lineage marker expression values were subjected to the landmark alignment procedure, and batch was included as a random effect in the linear mixed effects model analyses. This has been described in the methods section.

Reviewer #2, expert in food allergy (Remarks to the Author):

The topic of this manuscript, food allergy, belongs to the most dangerous epidemics around the world. The mechanisms by which infants get sensitized, and then develop towards sensitized but tolerant PST, or true food allergic PA, are largely not understood. Peanut is a paradigm of importance in the industrialized world with a lifelong risk of severe adverse events. The understanding how the tolerant state is transferred into the allergic is an unmet need. The presented work is novel and original. From a well characterized clinical sample, the HealthNuts cohort, 3 groups (each n=12) were recruited: In 36 infants being PST, PA or healthy (NA), carefully clinically characterized according to IgE values, SPT reactivity and reactivity upon oral challenge to peanut. From these cohorts, blood cells were isolated and examined by high dimensional mass cytometry-based single cell profiling. The frequency of cell species clusters was typed by manual gating, as well as by unsupervised analysis. Overall, the authors aim to explain the development of allergic sensitization with tolerance vs. clinical food allergy in the first year of life, while the results are a collection of observations in cellular phenotypes. The results do not allow a dynamic picture of how allergy develops and do not allow correlations with the elevated IgE in PA children, as the authors state themselves in line 235.

RESPONSE: We thank the reviewer for these comments. The goal of this study was to investigate the immune signatures that may contribute to the development of allergy over tolerance in peanut sensitised infants. This work is a comprehensive immune phenotyping study of early life food allergy and represents a framework for further investigation of these unique immune signatures. This has been clarified in the introduction (lines 63-66) and discussion of the manuscript (lines 210-212, lines 273-274).

Significant differences were found between T cells, B cells and plasmacytoid DCs:

- A) in PA children reduced frequency of the naïve CD4 T cell cluster and
- B) in PA children increases in peanut-specific memory CD4 T cells, CD45RA- memory phenotypes
- C) in PST in PST kids higher frequency of B cell cluster 2 (CD19++HLADR++ B cell) (compared to both groups PA or NA),
- D) in PST in PST kids higher frequency of plasmacytoid DCs (CD123+ CD11c- DCs) compared to NA.

Possibly the most important finding is that PST infants showed higher IL-2 levels with a functional role in tolerance, while for unknown reasons the cellular sources of IL2, IL-2+ cells were again comparable in naïve CD4 T cells in PA and PST infants.

On the other hand, PA showed higher potency for TNFalpha production by PBMC. This is really interesting as TNFalpha is uttermost important in anaphylaxis and prompts the question, at which time point the PBMCs were actually taken and cryo-conserved after the peanut challenges? Needs to be clarified in M&M. Under any circumstances cells were post challenge. This is pivotal as the observed signature in PA children may be a signature of the previous anaphylactic event during oral allergen challenge, instead of a clinical biomarker for the PA phenotype. Is it the chicken or the egg? Similarly, are the increases in peanut-activated CD4 T cells that display a memory phenotype in the PA group a memory of the recent allergen challenge? There is only one way to answer this: Defining a cellular signature before oral allergen challenge. To strengthen the conclusions, can the authors identify children at risk by this phenotype before doing an oral challenge?

RESPONSE: We agree, these are excellent overall questions that would entail separate research projects to be able to answer them rigorously in the future. For our specific research documented in the current manuscript, we feel we have made discoveries that will lay the groundwork towards future research. The question about oral allergen challenge is one we are currently studying using a broad 'omics' approach with another separate international cohort. For the HealthNuts cohort used in the current manuscript, blood was collected at clinic appointments 2 hours following peanut OFC. We have previously compared immune parameters in blood samples taken pre and post oral food challenge from food allergic infants, and have found no significant differences, including in the levels of inflammatory cytokines such as TNF α (in cell culture supernatants and plasma), between pre and post OFC samples. We refer you to the supplementary data of Neeland et al., JACI 2018 and Dang et al., Allergy 2013 for further details. We acknowledge this in our current study and have added this to the discussion section, line 262-268. We have also clarified the timing of blood collection in the methods section, line 287.

Apart from the IgE and cellular data, the determination of IgG4 could be very interesting and potentially discriminate the groups.

RESPONSE: We agree, measuring peanut-specific IgG4 in this cohort would be very interesting. This is a future goal of the HealthNuts cohort.

It is remarkable that the sensitivity of the cellular tests did not allow to determine IL4 and IL10, as these are really interesting biomarkers, but were they determined in the supernatants from stimulated samples?

RESPONSE: We did not measure IL-4 and IL-10 in cell culture supernatants in this study. Previous work in our laboratory has tested for IL-4 in cell culture supernatants from stimulated T cells and it was undetectable by cytometric bead array (Martino et al., Nature Comms 2018).

Finally, stimulation was done with pure peanut protein solution. In the light of the current discussion, the analysis of the extract and its compounds is missing as supplementary data. Which of the major allergens were contained, roasted or not, lipid content?

RESPONSE: We agree this is important information to add to the manuscript, thank you for the suggestion. The pure peanut protein was prepared in sterile PBS using peanut powder from Greer (cat. no.: XPF171D3A2.5). This peanut powder comes from raw peanuts (Virginia variety). As per correspondence with Greer technical specialists, the peanut powder is processed as follows: "The raw peanuts are ground and put into acetone through multiple soaks for defatting, followed by air and heat drying. They are then sieved to a particle size of <2.0mm" Assessment of Arah 1 and Arah 2 allergens (by Indoor Biotechnologies), revealed Arah 1 content: 71.03 and Arah 2 content: 78.43 (reported as microgram allergen per mL extract). The allergen content and the category number has been added to the methods section, line 296.

Reviewer #3, expert in pediatric food allergy (Remarks to the Author):

General Comments: In the study by Neeland and colleagues, the investigators utilized high dimensional mass cytometry-based immune profiling of resting and stimulated peripheral blood mononuclear cells (PBMCs) to define the circulating immune cell signatures associated with peanut-sensitized but tolerant infants (PSTs) vs. peanut-allergic infants (PA) vs. non-allergic healthy controls (NA) in the first year of life. Key features associated with PA in this cohort were the increased frequency of a unique subset of B cells (CD19++HLADR++) and over-production of TNF α following non-specific stimulation. Infants with PST were found to have a reduced frequency of (but hyper-responsive) naïve CD4 T cells and an increased frequency of plasmacytoid dendritic cells. Peanut-specific T cell responses were also evaluated and showed that infants with peanut allergy, but not PST, had higher proportions of memory peanut-specific CD4 T cells. The manuscript is well-written, and the methods clearly described.

Results showing the difference in cell populations might be easier appreciated in tabular form instead of writing them out as currently in the manuscript.

RESPONSE: Thank you for your positive comments. We have now expressed the proportions of each cell type in tabular format and added to the manuscript as Table 2.

While technically a very impressive study, given the small numbers of subjects studied and the variability among patient profiles seen, it is not clear how well this can be used to distinguish individual patients who are peanut allergic from those who are only sensitized.

RESPONSE: The goal of this study was to investigate the immune signatures that may contribute to the development of allergy over tolerance in peanut sensitised infants, that can help inform future tools for distinguishing these patients. This has been clarified in the introduction (lines 63-66) and discussion of the manuscript (lines 210-212, lines 273-274).

Specific Comments:

Lns 70-72: What criteria were used to select the various subsets of these cohorts from the HealthNuts population? Given the small numbers, this could significantly affect the outcome.

RESPONSE: The PA and PST participants were selected based on evidence of sensitisation as well as peanut OFC outcome. The healthy controls were selected based on the absence of sensitisation to any food as well as a negative peanut OFC. All participants selected in this study had a blood sample collected, and a cryopreserved PBMC number greater than 8×10^6 . Detailed demographics of the selected cohort are presented in Table 1.

Lns 80-81: Is this sentence necessary? By definition the PA and PST subsets had to have evidence of IgE and the normal controls could not have positive skin tests or peanut-specific IgE.

RESPONSE: Thank you for this feedback, this sentence has been removed from the manuscript.

Lns 85-137: The percentages of different cell types in this section would be easier to compare if presented in tabular form.

RESPONSE: We have now expressed the proportions of each cell type in tabular format and added to the manuscript as Table 2.

Lns 294-295: Was the proportion of live cell types following thawing the same as the proportion of fresh PBMC cell types, especially APCs?

RESPONSE: Unfortunately, it was not possible to perform these experiments on fresh cells. Several previous studies have compared fresh and thawed PBMC, revealing no significant effect of cryopreservation in phenotype or enumeration of innate and adaptive immune cells (Weinberg et al., Clin Diagn Lab Immunol 2000, Veluchamy et al., Scientific Reports 2017, Ramachandran et al. Cells 2012, Mandl et al., PLoS One 2014).

Lns 388-389: Given the number of comparisons made, is an FDR set at $p < 0.1$ adequate?

RESPONSE: Thank you for raising this valid question. We decided on the 10% cutoff before starting the analysis and we maintained this plan throughout. For consistency, we decided to control the false discovery rate to be no more than 10% for all analyses, including the specific line the reviewer is referring to, where there are only 13 tests. We believe it is appropriate that there is consistency in interpretation of findings and it would not be appropriate for the level of control to be a function of the number of tests. For clarification, we have added the total number of tests performed for each analysis throughout the methods section.

REVIEWERS' COMMENTS:

Reviewer #1 (Remarks to the Author):

The revised manuscript is much improved. The authors have addressed all my critiques, I have no further comments.

Reviewer #2 (Remarks to the Author):

The authors have to a great part satisfactorily addressed my comments.

Still, I suggest to add a paragraph "limitations of the study": It is important to underline that the paper is a snap-shot (after previous clinical reactivity to peanut PLUS cellular fingerprint 2h after an allergen challenge), that hence does not allow conclusions on the dynamic "development" of peanut sensitization and clinical allergy in the first year.

The last sentence in abstract on development is too speculative. Replace "drive development of" by "characterize". Also the last sentence 272-274 promises too much.

Please spell Ara h 1 acc. to nomenclature.

I will be happy to see in future papers the investigation of IgG4, and I would have liked to see the data of IL-4/13 and IL-10 for completeness.

I am also looking forward to see data using cells before a peanut challenge in future work.

Reviewer #3 (Remarks to the Author):

The authors have adequately addressed my concerns.

Dear Dr. Bondar,

Please see below for our responses to Reviewer 2.

REVIEWERS' COMMENTS:

Reviewer #1 (Remarks to the Author):

The revised manuscript is much improved. The authors have addressed all my critiques, I have no further comments.

Reviewer #2 (Remarks to the Author):

The authors have to a great part satisfactorily addressed my comments.

Still, I suggest to add a paragraph "limitations of the study": It is important to underline that the paper is a snap-shot (after previous clinical reactivity to peanut PLUS cellular fingerprint 2h after an allergen challenge), that hence does not allow conclusions on the dynamic "development" of peanut sensitization and clinical allergy in the first year.

As suggested by the reviewer, we have added the following to the discussion:

"We acknowledge this study has several limitations. Whilst comprehensive for the well characterized lymphoid and myeloid markers, a limitation of our antibody panel is that it does not include surface markers for innate-like and unconventional T cell populations, or subsets of immunoglobulin switched/non-switched B cells, which may be relevant for the development of an allergic immune response. It should also be considered when interpreting our findings that, as blood collection was performed following clinical testing, a positive peanut OFC could influence some of the immune parameters investigated in this study. As such, the results presented represent a snapshot of the immune response following *in vivo* allergen exposure. However, we have previously reported no differences in cellular activation or plasma cytokine production in food allergic infants who had a blood sample taken on a non-OFC day versus an active OFC day. It will be interesting to determine if these signatures can be identified prior to allergen challenge, and we acknowledge these are important areas for future investigation into the development of peanut sensitization and allergy in the first year of life."

The last sentence in abstract on development is too speculative. Replace "drive development of" by "characterize". Also the last sentence 272-274 promises too much.

These sentences have been removed.

Please spell Ara h 1 acc. to nomenclature.

This has been edited in the methods section.

I will be happy to see in future papers the investigation of IgG4, and I would have liked to see the data of IL-4/13 and IL-10 for completeness. I am also looking forward to see data using cells before a peanut challenge in future work.

Reviewer #3 (Remarks to the Author):

The authors have adequately addressed my concerns.